# The Role of Immunonutrition in Patients Undergoing Pancreaticoduodenectomy

**DOI:** 10.3390/nu12092547

**Published:** 2020-08-23

**Authors:** Beata Jabłońska, Sławomir Mrowiec

**Affiliations:** Department of Digestive Tract Surgery, Medical University of Silesia, Medyków 14 St., 40-752 Katowice, Poland; mrowasm@poczta.onet.pl

**Keywords:** immunonutrition, pancreaticoduodenectomy, glutamine, arginine, omega-3 fatty acids, nucleotides

## Abstract

Pancreaticoduodenectomy (PD) is one of the most difficult and complex surgical procedures in abdominal surgery. Malnutrition and immune dysfunction in patients with pancreatic cancer (PC) may lead to a higher risk of postoperative infectious complications. Although immunonutrition (IN) is recommended for enhanced recovery after surgery (ERAS) in patients undergoing PD for 5–7 days perioperatively, its role in patients undergoing pancreatectomy is still unclear and controversial. It is known that the proper surgical technique is very important in order to reduce a risk of postoperative complications, such as a pancreatic fistula, and to improve disease-free survival in patients following PD. However, it has been proven that IN decreases the risk of infectious complications, and shortens hospital stays in patients undergoing PD. This is a result of the impact on altered inflammatory responses in patients with cancer. Both enteral and parenteral, as well as preoperative and postoperative IN, using various nutrients, such as glutamine, arginine, omega-3 fatty acids and nucleotides, is administered. The most frequently used preoperative oral supplementation is recommended. The aim of this paper is to present the indications and benefits of IN in patients undergoing PD.

## 1. Introduction

Pancreaticoduodenectomy (PD) is one of the most complex surgical procedures in gastrointestinal surgery. Most frequently, it is indicated for patients with malignant tumors of the pancreatic head, the distal common bile duct, and periampullary tumors. Additionally, some benign pancreatic and periammpullary diseases, such as inflammatory tumors within pancreatic head in the course of chronic pancreatitis or other benign diseases, are indications for PD. Regardless of surgery indications, PD is associated with a risk of postoperative complications. The morbidity rate after PD has (30–40%) decreased [1,2]. However, the perioperative mortality rate still ranges 0–5% [1]. Postoperative pancreatic fistula (POPF) remains the most common complication after PD, and this complication significantly prolongs the duration of hospitalization and leads to high medical costs [1].

The ways to reduce the incidences of postoperative complications after PD are still being sought. Careful surgical techniques, with the choice of the optimal reconstruction method, is most crucial to improve postoperative outcomes [3]. However, nutritional support in patients undergoing PD is also very important, because a significant number of these patients are malnourished due to the basic disease. Additionally, surgical trauma leads to immune system suppression. The surgical injury, as well as the malnutrition, can provide poor postoperative outcomes. Therefore, various pre- and postoperative nutritional interventions, including oral, enteral and parenteral formulas, are used in patients following PD [2]. Immunomodulating nutrition (IMN) or immunonutrition (IN) is the kind of nutritional support that uses special nutrients modulating the inflammatory postinjury response and counteracts postoperative immune impairment. There are four main imunomodulating substrates used in IN: glutamine (Gln), arginine (Arg), omega-3 polyunsaturated fatty acids (PUFAs) and nucleotides [4]. The first studies on IN involved experimental animal models. The authors noted the failure of the gut barrier function and integrity with subsequent bacterial translocation after injury. Therefore, in first experimental studies, injured animals were fed enterally rather than parenterally due to the protective effect of gut feeding on bacterial translocation [4]. The first clinical studies on IN were performed by Marco Braga and his surgical resident Luca Giannotti in the 1990s. In 1996, these authors reported that immunonutrients should be given before surgery in order to obtain adequate concentrations at the time of surgical stress, because the impairment of the host defense mechanisms occurred immediately after surgery [4,5]. In another study, the authors demonstrated a significant decrease in postoperative infectious complications and lengths of hospital stays in patients with gastrointestinal cancers fed before and after surgery with a diet supplemented with arginine, omega-3 fatty acids and nucleotides, in comparison to patients fed with a standard enteral formula. A decrease in morbidity rates and hospital stays can significantly reduce hospital costs [4,6].

Since then, many reports, metaanalyses and randomized prospective studies have been published. However, the role of IN in patients undergoing PD is still unclear and doubtful in some authors’ opinions, and the reports on IN in PD are contradictory. The aim of this paper is to review recent publications regarding use of IN in patients undergoing PD.

## 2. Enhanced Recovery after Surgery (ERAS) Guidelines

Firstly, the most recent ERAS guidelines on clinical nutrition in surgery do not strongly recommend IN in all patients undergoing major surgery (including PD). According to the 11th recommendation [7], parenteral glutamine supplementation may be considered in patients who cannot be fed adequately enterally and, therefore, require parenteral nutrition (PN) (grade of recommendation: B). Currently, there is no clear recommendation regarding the supplementation of oral glutamine [7]. According to the 12th recommendation [7], postoperative parenteral nutrition, including omega-3-fatty acids, should be considered only in patients who cannot be adequately fed enterally and, therefore, require parenteral nutrition (grade of recommendation: B) [7]. According to the 13th recommendation [7], the peri- or at least postoperative administration of specific formulae enriched with immunonutrients (arginine, omega-3-fatty acids, ribo-nucleotides) should be given in malnourished patients undergoing major cancer surgery (grade of recommendation: B) [7]. Currently, there is no clear evidence for the use of these formulas enriched with immunonutrients vs. standard oral nutritional supplements exclusively in the preoperative period. Currently, no clear recommendation can be given regarding the parenteral or enteral supplementation of arginine as a single substance [7]. It should be emphasized that, according to 17th strong (grade A) recommendation [7], preoperatively, oral nutritional supplements should be given to all malnourished cancer and high-risk patients undergoing major abdominal surgery. Moreover [7], the 18th recommendation [7] prefers the preoperative (5–7 days before surgery) administration of oral IN supplements, including arginine, omega-3 fatty acids and nucleotides [7].

## 3. The Use of IN in Patients Undergoing PD in the Recent World Literature

There are many publications regarding the role of IN in surgical patients in the world literature. Firstly, we found all papers on IN in surgery and oncology in the PubMed database [8,9,10,11,12,13,14,15,16,17,18,19,20]. In the next approach, we selected and reviewed articles on the role of IN in major gastrointestinal surgery. Finally, only the literature regarding IN in patients undergoing PD was widely discussed in this review paper. In this patient group, the reports are not clear. The search strategy of the literature review (step by step) is presented in Table 1.

One of the most important studies regarding the role of IN in patients undergoing PD is the study of Gianotti et al. [21], published in 2000. This is a prospective, randomized trial carried out in 212 patients undergoing PD. Patients were randomized to receive a standard enteral formula (standard group including 73 patients) or an enteral IN formula enriched with arginine, omega-3 fatty acids, and RNA (Impact; Novartis Nutrition, Bern, Switzerland) (immunonutrition group including 71 patients), or total parenteral nutrition (parenteral group including 68 patients). The administration of enteral diet started 6 h after surgery at 10 mL/h rate. The velocity was progressively increased by 20 mL/day until reaching full nutritional goal (25 kcal/kg/day). Enteral or parenteral infusion was continued until the patient’s oral intake was approximately 800 kcal/d. The authors noted a significantly lower rate of postoperative complications and the shorter length of hospital stay in the IN group (33.8%/15.1 days) compared to the standard (43.8%/17.0 days) or parenteral groups (58.8%/18.8 days), respectively [21].

In 2019, Miyauchi et al. [22] published a prospective, randomized clinical trial. The authors compared perioperative and preoperative IN assessing cell-mediated immunity and the infection rate in patients following PD. The authors administered oral IN (arginine, ω-3 fatty acids, and dietary nucleotides) (Impact; Nestle Health Science Co., Ltd., Kobe, Japan) and enteral IN to 30 patients before and after surgery (perioperative group); 30 patients received the same enriched formula before surgery and standard enteral nutrition following surgery (preoperative group). The patients received oral supplementation (1000 kcal/d) for 5 days before surgery. A total of 1000 mL of oral Impact contained 12.8 g of arginine, 3.4 g of omega-3-fatty acids and 1.29 g of nucleotides. All patients were hospitalized by 6 days before surgery. Postoperative enteral feeding with oral Impact in the perioperative group or standard formula in the preoperative group started about 12 h after surgery. A gastrostomy catheter, placed into the jejunum intraoperatively, was used for enteral feeding in both groups. Enteral feeding was started on postoperative day 1 at 20 mL/h and was increased progressively by 20 mL/day until the full nutritional goal (25 kcal/kg/day) was reached. All patients were allowed to drink water on postoperative day 2, and oral food intake was allowed from postoperative days 3–5, according to clinical conditions. The enteral feeding was continued until oral intake was approximately 800 kcal/day. The authors concluded that there were no additional effects of perioperative, compared with preoperative IN, on postoperative immunity and infectious complications in patients following PD [22].

In 2016, Silvestri et al. [23] analyzed the role of preoperative IN on postoperative outcomes in patients undergoing PD. In this study, 54 well-nourished patients undergoing PD received preoperative oral IN (Impact; Nestle, Italia)) (Impact composition: Arginine 1.8 g, omega-3-fatty acids 0.6 g, nucleotides 0.2 g) for 5 days before surgery at a dose of 750 mL/day (three packs). From the first postoperative day, all patients were treated with total parenteral nutrition at a dose of 30 kcal/kg/d and 0.2–0.3 g of Nitrogen per kg/day. Oral diet was introduced progressively from the fifth postoperative day (in the absence of POPF). These patients were compared with a control group receiving preoperative standard oral diet. The authors demonstrated no statistical differences in mortality (2.1% in each groups) and overall morbidity rate (41.6% vs. 47.9%) between both groups. The authors noted a statistically lower rate of infectious complications (22.9% vs. 43.7%, *p* = 0.034) and shorter hospital stay (18.3 ± 6.8 days vs. 21.7 ± 8.3, *p* = 0.035) in the IN group [23].

There are some articles on the mechanisms of IN on reducing infectious complications following surgery in the literature. Suzuki et al. [24], in a prospective randomized study, determined whether IN influenced the immunological parameters as follows: cell-mediated immunity, differentiation of T helper type 1 (Th1) and Th2 cells, interleukin (IL)-17-producing CD4(+) helper T (Th17) cell response and infectious complication rate after PD. In this study, the authors divided patients undergoing PD into three groups, counting 10 patients: the perioperative group received IN (Oral Impact; Ajinomoto Pharma Co., Ltd., Tokyo, Japan) for 5 days (1000 kcal/day) before surgery, which was prolonged after surgery by enteral infusion; the postoperative group received an early postoperative enteral infusion of the same enriched formula with no artificial nutrition before surgery; the control group received total parenteral nutrition postoperatively. Enteral feeding started at 12–48 h after surgery at 10 mL/h rate. The velocity was increased progressively by 20 mL/day until 25 kcal/kg/day was reached. Oral food intake was allowed on postoperative day 7. The authors concluded that perioperative IN reduced the rate of infectious complications and stress-induced immunosuppression after PD. In the authors’ opinion, the modulation of Th1/Th2 differentiation and Th17 response may play important roles in this immunologic effect [24].

Aida et al. [25], in a prospective randomized study, assessed the impact of preoperative IN on postoperative complications and the participation of prostaglandin E2 (PGE2) on T-cell differentiation in patients undergoing PD. The patients were divided into two groups according to nutritional intervention: 25 patients with preoperative IN (Oral Impact; Ajinomoto Pharma Co., Ltd., Tokyo, Japan) at a dose of 1000 kcal/day and 25 patients in the control group with standard nutrition administered for 5 days before surgery (2000 kcal/day). Both groups received the same postoperative nutrition. A gastrostomy catheter, inserted into the jejunum intraoperatively, was used for enteral feeding in both groups. Enteral nutrition was started on the first postoperative day at a of dose 20 mL/h and it was increased progressively by 20 mL/day. Oral food intake was started on the fifth postoperative day with a gradual decrease in enteral feeding. The study showed that the infectious complication rate and severity of complications according to the Clavien–Dindo classification were lesser in the IN group compared to the control group. Additionally, significant differences in the levels of immunological parameters were noted between two groups. In the IN groups, mRNA expression levels of T-bet, serum eicosapentaenoic acid and eicosapentaenoic acid/arachidonic acid ratios were significantly greater and the levels of plasma prostaglandin E2 were significantly lesser compared to the control group [25].

Gade et al. [26] assessed the impact of oral IN (Oral Impact Powder; Nestle, Vevey, Switzerland) used for 7 days before surgery for pancreatic cancer on postoperative complications and length of hospital stay in a randomized controlled trial on 35 patients, including 19 (54%) patients with IN oral intervention. The intervention with IN was ended on the day before surgery. The IN group was compared with the control patients receiving the standard diet. There were not any statistical differences in postoperative complications and length of hospital stay between analyzed groups in this study [26].

In the randomized clinical trial performed by Hamza et al. [27], 37 patients, undergoing PD for periampullary cancer, were randomized, including 17 patients with IN intervention (Impact; Novartis Medical Nutrition, Horsham, West Sussex, UK) and 20 patients receiving the standard isocaloric isonitrogenous diet (Fresubin; Fresenius Kabi Ltd., Runcorn, UK). Patients received three cartons (200 mL per carton) of either feed per day for 14 days before surgery. No patient received total parenteral nutrition in the postoperative period of the trial. Postoperatively, feeding was administered via nasojejunal tube within 24 h after surgery at a rate of 25 mL/h with a gradual increase in the rate to achieve a rate of 25 kcal/kg/day by the third or fourth postoperative day. The enteral feeding was continued for a minimum of 7 days postoperatively. The patients were fed for 14 days preoperatively and 7 days postoperatively. The authors noted a favorable modulation in the inflammatory response and improvement of systemic immunity in the IN group [27].

Ashida et al. [28] analyzed the impact of IN on the incidence of hypercytokinemia after PD in a double-blinded randomized controlled trial. The study included 20 patients: 11 patients received preoperative IN (enteral diets enriched in eicosapentaenoic acid (EPA)) (Prosure; Abbot Japan Co., Ltd., Tokyo, Japan) in addition to 1200 kcal of regular food, and 9 patients were fed using an isocaloric isonitrogenous standard nutrition (600 kcal/day) without EPA (Procure Z; Nisshin Oillio Group, Ltd., Tokyo, Japan) for 7 days before surgery due to periampullary cancer. This study did not demonstrate the significant impact of preoperative IN on the rates of postoperative hypercytokinemia or infectious complications in 50 patients following PD [28].

Shirakawa et al. [29] analyzed the impact of preoperative oral IN on postoperative outcome in patients undergoing PD. The study involved 25 patients divided into two groups: 18 patients receiving oral IN (Impact; Ajinomoto Pharma) 5 days before surgery (750 mL/day: three packs of 250 mL per day) and 15 patients without IN received a standard oral diet. The authors reported a significantly lower rate of wound infection in patients receiving IN compared to the control group (0% vs. 30.8%, *p* = 0.012) [29].

It is important to know which patients may benefit from IN in order to select a group of patients who can be recommended IN as standard. In 2020, Tumas et al. [30] studied 70 patients undergoing PD for pancreatic tumors randomized into IN vs. control groups and stratified them according to final histological diagnosis and nutritional status. The IN group received 5 days of preoperative IN (L-arginine 6.04 g/day and polyunsaturated fat 4 g/day) in addition to the usual preoperative nutritional management. The control group received routine preoperative nutritional management only. All patients received infusions of 200 mL of 5% glucose solution on the morning of surgery. On postoperative days 1–3, patients received normocaloric enteral formula in an increasing dose, gradually replaced by oral nutrition on postoperative days 4–5, according to the clinical conditions. The authors assessed the impact of IN on the overall complication rate and the rate of severe and/or multiple complications in patients with pancreatic tumors stratified according to final histological diagnosis patients with pancreatic ductal adenocarcinoma (PDAC) vs. other tumors and nutritional status. In this study, there were no significant differences in the overall complication rates in IN vs. control, patients with malnutrition vs. no malnutrition or PDAC vs. other pancreatic tumors. However, significant differences in the rates of severe and/or multiple complications in IN vs. control groups and in PDAC patients divided according to IN were demonstrated [30].

Furakawa et al. [31] assessed the impact of oral IN on infectious complications in low skeletal muscle mass patients after PD, in a retrospective, consecutive cohort study. Skeletal muscle mass was assessed using preoperative computed tomography images in 298 consecutive patients who underwent PD. Risk factors for postoperative infectious complications and the impact of preoperative IN on low SMI patients undergoing PD were analyzed. Of the 298 patients, 91 patients received preoperative IN, containing omega-3 fatty acids, arginine, and nucleotides (Oral Impact; Nestle Health Science Co., Ltd., Kobe, Japan), for 5 days before surgery, in addition to a 50% reduction in the amount of regular dietary intake (1000 kcal/day). A total of 1000 mL of oral Impact contained 2.0 g of EPA, 12.8 g of arginine, and 1.29 g of nucleotides. Patients, not receiving IN, consumed a regular normocaloric diet before surgery (2000 kcal/day). All patients receiving preoperative IN were hospitalized for 6 days prior to surgery. Postoperative enteral feeding, using the standard formula, was started on the first postoperative day via gastrostomy catheter at a 20 mL/h rate and increased progressively by 20 mL/day, in order to achieve 25 kcal/kg/day. All patients were allowed to drink water on the second postoperative day. The authors compared the impact of IN on infectious complications between two groups: high and low skeletal muscle mass index (SMI). In high SMI patients, the rate of postoperative infectious complications was significantly lower in those who received IN than in those who did not receive IN (31.9% vs. 46.1%, respectively; OR, 1.82; *p* = 0.045). Similar results were reported in low SMI patients (26.3% vs. 83.6%, respectively; OR, 14.31; *p* < 0.001), but Odds Ratio (OR) was significantly (seven times) higher in low vs. high SMI patients. The authors observed a significantly lower plasma IL-6 level in low SMI patients receiving IN compared to those who did not receive IN. Therefore, the authors concluded, that there was a stronger association with reduced infectious complications in patients who had low SMI and received IN. Additionally, the authors analyzed the association between main pancreatic duct diameter and IN. This study revealed that there was no association between main pancreatic duct diameter and IN. There was no influence of main pancreatic duct diameter on outcomes of IN. This study has shown that patients with low SMI are candidates for IN before PD. This is a very important conclusion, because sarcopenia occurs in patients undergoing PD. It may be caused by ageing, cancer, malnutrition, sepsis, and some chronic diseases [31].

We would like to cite and discuss two recent meta-analysis studies regarding the role of IN in patients undergoing PD. In 2019, Guan et al. [17] analyzed 4 randomized clinical trials including 299 patients. This analysis showed that IN significantly reduced the incidence of postoperative infectious complications (RR 0.58, 95% CI 0.37–0.92; *p* = 0.02) and shortened the length of hospital stay (MD −1.79, 95% CI −3.40 to 0.18; *p* = 0.03). In this study, there were no significant differences in the incidence of overall postoperative complications (RR 0.81, 95% CI 0.62–1.05; *p* = 0.11), non-infectious complications (RR 0.94, 95% CI 0.69–1.28; *p* = 0.70) and postoperative mortality (RR 2.43, 95% CI 0.37–16.10; *p* = 0.36). Therefore, the authors concluded that IN significantly decreased the rate of infectious postoperative complications and duration of hospitalization, but did not influence the rate of overall postoperative complications and non-infectious complications [17].

In 2020, Takagi et al. [16] published a systematic literature review of randomized controlled trials in order to identify the studies investigating the impact of IN on outcomes in patients following PD. This analysis included five studies and demonstrated a lower incidence of overall complications (RR 0.74; 95% confidence interval (CI) 0.58, 0.94; *p* = 0.01; I^2^ = 0%) and infectious complications (RR 0.60; 95% CI 0.42, 0.84; *p* = 0.003; I^2^ = 0%). However, the incidence of major complications (RR 0.68; 95% CI 0.41, 1.12; *p* = 0.13), mortality (RR 0.79; 95% CI 0.16, 3.99; *p* = 0.78), postoperative pancreatic fistula (RR 0.92, 95% CI 0.59, 1.46; *p* = 0.74), and delayed gastric emptying (DGE) (RR 1.09; 95% CI 0.55, 2.15; *p* = 0.81) were comparable in patients receiving IN and standard diet [16].

We also reviewed articles regarding parenteral IN in patients undergoing PD. The parenteral administration of immunonutrients is less frequent than the oral one. The glutamine is commonly used in parenteral IN [17]. Sungho et al. [32] analyzed the influence of Gln on the surgical outcome in patients undergoing PD for periampullary tumors. This prospective, randomized, double-blind, and controlled clinical trial included 60 patients: 32 had been in the Gln group and 28 in the control group. The Gln and control groups received isonitrogenous amino acid, with a 0.2 g/kg per day Gln regimen administered to the Gln group. The median length of the postoperative hospital stay, and the postoperative nutritional and chemical parameters were comparable in two groups. Additionally, the overall and PD-related complication rates of the Gln group (37.5% and 25.0%) and the control group (28.6% and 14.3%) were comparable. The authors concluded that there was no significant benefit of Gln low-dose parenteral administration in patients undergoing PD [32].

The impact of parenterally administered Gln on the postoperative outcome in major gastrointestinal surgery is still controversial. To confirm this, we would like to cite the recent (2020) article regarding the parenteral administration of Gln in patients undergoing the other major gastrointestinal surgery, gastrectomy. Wu et al. [33] analyzed 1950 patients, among whom 522 (26.8%) received parenteral glutamine supplementation (glutamine dose ranging from 0.05 to 0.49 g/kg/day) in a single-center cohort study. The authors did not note any impact of Gln on the surgical outcome, but the significant improvement of serum albumin after surgery was observed [33]. Therefore, the authors concluded that the use of IN in patients undergoing PD could prevent the incidence of overall and infectious postoperative complications, but it did not influence major complications, mortality, and PD-specific complications, such as POPF and DGE [16]. A summary of recent articles regarding the role of IN in patients undergoing PD is presented in Table 2.

A previous prospective randomized controlled trial conducted by Jeh et al. [34] on 70 patients undergoing major gastrointestinal surgery showed that Gln supplementation significantly decreased postoperative inflammation and immunological and nutritional depressions in operated patients. The patients had received Gln parenterally in the perioperative period: from 1 day before surgery to the sixth day after surgery, for 7 days in summary [34].

## 4. Immunomodulating Mechanisms of Nutrients Used in IN

We would like to present the immunomodulating mechanisms of the abovementioned nutrients. Most authors have reported a decreased risk of infectious complications in patients following surgery. In order to show the “anti-infectious” role of IN, the mechanism of postoperative infections should be explained. In patients undergoing PD, bacterial flora from the gut, especially *Enterococci* and *Escherichia coli*, translocate into the mesenteric lymph nodes or blood, where the most postoperative infections are observed. Some factors in the perioperative period can facilitate this bacterial translocation, such as a reduction in jaundice, postoperative intestinal motility, the use of antibiotics resulting in small bowel bacterial overgrowth, the deterioration of the mucosal barrier due to malnutrition, intestinal manipulation, and parenteral nutrition. Therefore, immunonutrients should influence the immunological response in patients following surgery [29,35].

Stress caused by PD induces systemic inflammation with the production of inflammatory mediators, such as IL-1 beta, IL-6 and TNF-alfa. The excessive production of these cytokines, especially IL-6, is associated with an increased risk of postoperative infectious complications following major abdominal surgery, including PD. Multiple studies have shown a significant postoperative hypercytokinemia in patients following PD. In order to decrease the risk of postoperative infectious complications in patients undergoing PD, the production of these cytokines should be decreased. IN may suppress postoperative hypercytokinemia and reduce the rate of infectious complications following PD, which is associated with a shorter duration of hospitalization [28]. It has been shown that PD is one of the most stressful surgeries. The proof of this theory is that the level of plasma IL-6 after PD was higher than after gastric and colorectal surgery. Furthermore, stress-induced immunosuppression was greater after PD, compared to gastric and colorectal surgery [22]. IN modulates the inflammatory response and the production of inflammatory mediators in patients undergoing PD. Additionally, it favorably modulates systemic and mucosal immunity [27]. It is important that IN has the most benefits when it is administered a minimum of 5 days before PD. It is associated with the fact that preoperative IN for a minimum 5 days is necessary to show its immunometabolic results, following PD [25]. In patients undergoing PD, a reduction in the host response and the immunity is noted. They are facilitated by low caloric intake and by intestinal bacterial translocation that are observed in patients following PD. The risk factors facilitating immunological disturbances are: alteration of postoperative intestinal motility and loss of mucosal barrier function. The impaired immunological activity leads to an increased risk of postoperative infectious complications (wound infection, pneumonia, infection of the urinary tract, an infected pancreatic fistula, enteritis, sepsis) and this is associated with a prolonged hospital stay. Prolonged duration of hospitalization is associated with increased hospital costs [22,23]. Surgery leads to a transient immunosuppression and potential alterations in gastrointestinal tract function. In patients following surgical injury, an excessive inflammatory response and the paralysis of cell-mediated immunity may be observed. It can lead to postoperative infectious complications. The gastrointestinal tract has been called “the undrained abscess” of multiple organ failure. Postoperative gut barrier failure is associated with an increase in intestinal permeability and bacterial translocation. It leads to the activation of immunocompetent cells within the gut wall and associated lymph nodes: mucosa-associated lymphoid tissue (MALT) and gut-associated lymphoid tissue (GALT). The combination of increased barrier permeability and gut inflammation may potentially result in remote organ dysfunction caused by the interaction between polymorphonuclear leukocytes and the endothelial lining [36]. Obstructive jaundice, as a result of tumor infiltration or compression on the common bile duct due to a basic disease, is frequently noted in patients undergoing PD. It has been proven that it has been associated with an impaired immune function, including both the systemic and local defense. The gut barrier dysfunction and endotoxemia were noted in these patients. An increased risk of infectious and septic postoperative complications has been seen in patients with obstructive jaundice. Biliary obstruction results in an increase in intestinal permeability and the upregulation of HLA-DR expression on enterocytes and GALT, suggesting an immune activation [36].

It was reported that perioperative IN significantly reduced the risk of postoperative infectious complications by 50% [37]. It should be emphasized that, although WHO and Centers for Disease Control and Prevention guidelines for the prevention of surgical site infection did not refer to the significance of nutrition therapy in previous versions, the 2016 WHO guidelines listed IN as a method contributing to the prevention of surgical site infection [38]. According to the second recommendation, multiple nutrient-enhanced formulas can be used to prevent surgical site infections in adult patients undergoing major surgery. According to authors of these recommendations, a careful selection of candidates for nutritional support is needed, because the use of enhanced nutrition support is expensive and requires additional work for hospital staff [38].

The route of IN is also very important because it influences the host defense mechanism. Comparing enteral and parenteral routes, enteral nutrition (EN) is better than total parenteral nutrition (TPN) [37]. EN preserves and TPN reduces gut-associated lymphoid tissue cell number, gut immunoglobulin A (IgA) level, respiratory tract IgA level, hepatic mononuclear cell number, resident macrophage number, and exudative neutrophil number [37]. EN significantly, and TPN poorly, influences resistance against viruses and bacteria, intracellular signaling activation, cytokine production, and nuclear factor-κB activation. It is important that survival in portal bacteriemia and bacterial peritonitis is good for EN and poor for TPN [37].

Utsumi et al. [39] reported that the preoperative nutritional assessment using the Controlling Nutritional Status (CONUT) Score may predict the pancreatic fistula after PD. The CONUT score is a tool used to assess nutritional status. It takes into account the following laboratory parameters: serum albumin level (indicating protein reserve), total cholesterol level (indicating calorie depletion), and total lymphocyte count (indicating loss of immune defense caused by immune malnutrition). The authors concluded that patients with high CONUT scores were at high risk for POPF. According to them, preoperative immunonutrition might help reduce the POPF risk in these patients through modulating impaired immunity in them [39]. It regards the clinically relevant POPF classified as grade B and C, according to the International Study Group on Pancreatic Fistula classification [40]. POPF with an elevated inflammatory response observed in blood examination and following the intravenous antibiotic administration was defined as grade B POPF, caused by infection. In the case of organ failure occurring, a grade C POPF is recognized. Therefore, immunomodulating influences not only typical infectious complications, but can also decrease the risk of POPF—the common complication following PD (incidence 11.4–64.3% of PD) [39].

## 5. Glutamine (Gln)

Glutamine is a conditionally essential amino acid. In conditions of severe surgical stress, Gln production may not achieve the amount needed, leading to glutamine deficiency [37]. Glutamine is a common nitrogen donor for healing tissues damaged by surgery. There are many studies showing the benefits of oral or enteral glutamine supplementation, in order to improve the quality of life of cancer patients, that are associated with better nutrition, but also decreased mucosal damage (mucositis, stomatitis, pharyngitis, esophagitis and enteritis) [41]. It is a preferred fuel for rapidly proliferating cells, such as gut mucosal cells (enterocytes), lymphocytes and neutrophils [36,37]. It also improves neutrophil, lymphocyte and intestinal function. This amino acid also maintains a normal GALT function and respiratory immunity [36]. It is a material for the synthesis of glutathione, a potent intrinsic antioxidant, and enhances heat shock protein expression [37]. Therefore, it is useful in order to modulate impaired immune response and the dysfunction of the intestinal barrier in patients following PD. Additionally, Gln is a precursor for protein, nucleotid and nucleic acid synthesis and regulates various cellular pathways and related functions. Therefore, Gln is important for intestinal integrity and function, proper immunologic response and antioxidative balance. It should be known that Gln concentration is decreased in patients in catabolism and stress. The insufficient endogenous availability of Gln may impair outcomes in critically ill patients. In patients following major surgery, Gln is observed due to increased metabolic demand. The liver, kidney and gastrointestinal tract are critical organs [42]. Gln plays multiple important immunological roles. It influences cellular activity—as a precursor of purine and pyrimidine compounds and glutathione. It influences nitric oxide metabolism in interaction with arginine (second immunonutrient), regulates cell maturation, stimulates the production of heat shot proteins (HSP), increases the cytotoxicity of tumor necrosis factor (TNF) alpha and activates kinases that are responsible for extracellular communication [43]. Additionally, it influences lymphocyte activity, such as: the stimulation of Concanavalin A (Con-A) and phytohemagglutinin (PHA)-induced proliferation, the activation of CD 25, CD 71, CD 45RO expression, the stimulation of Interferon (INF)–gamma secretion, the stimulation of natural killer (NK) cells, the inhibition of apoptosis, the stimulation of gut-associated lymphoid tissue (GALT) and the increase in NK population in the spleen. In addition, Gln influences the activity of monocytes: the stimulation of RNA synthesis, the increase in IL-1 secretion, the stimulation of phagocytosis, the stimulation of antigen presentation and the stimulation of monocyte maturation [44]. Rodent studies conducted by Kles et al. [38] on the ischemia-reperfusion model (IR) model, proved that Gln was better absorbed than glucose in the intestinal villi, which was further increased with increased amino acid supply [45]. It has been proven that Gln with arginine stimulates wound healing. Williams et al. [46] observed an increase in the concentration of hydroxyproline of 19% and an increased percentage of healed wounds in patients who had received glutamine and arginine [46].

It should be noted that parenteral nutrition formula containing glutamine reverses the lack of enteral nutrition-induced GALT atrophy. Therefore, it is worth adding glutamine to the parenteral mixture in patients who cannot be fed enterally in order to prevent immune gut dysfunction and the atrophy of enterocytes in patients fed only parenterally [37].

The positive role of Gln on prognosis has been proven in critically ill patients [36,47,48,49,50]. However, data regarding the use of Gln in these patients are also contradictory [39,47,48,49,50,51,52,53,54,55,56], because not all studies have confirmed this theory. Some authors have proven that high doses of Gln are associated with an increased mortality rate in critically ill patients [52,53,54,55,56]. According to them there are not recommendations for routine use of Gln in all patients of intensive care units [51,52,53,54,55]. Therefore, Gln use in intensive care patients is also controversial [57,58].

## 6. Other Immunonutrients (Arginine, PUFAs, Nucleotides)

Arginine, the second immunonutrient, is a semi-essential amino acid for catabolism, playing an important role in protein synthesis [29,59]. It promotes T cells and increases their activity and stimulates the phagocytosis of neutrophils. Yeh et al. [60], in the study conducted on septic rats, reported that Arg reduced the production of the following inflammatory mediators: interleukin (IL)-1 beta, tumor necrosis factor alpha (TNF-a) and IL-6, IL-18 at the site of tissue injury. It also stimulates tissue growth after infection [29,36,60,61]. Arg improves wound healing and regeneration and modulates inflammation and the immune response [23]. Vidal-Casariego et al. [62] noted that the administration of Arg-enriched enteral nutrition led to a significant reduction in fistulas and hospital stays in patients undergoing surgery for head and neck cancer [60]. Additionally, Arg induces the secretion of hormones: pituitary growth hormone, insulin-like growth factor IgF-1, insulin, vasopressin, catecholamines and somatostatin. It inhibits NF-*κ*B translocation and blocks adhesion molecules and inhibits lipid peroxidation [36,63]. Arg is also a substrate of nitric oxide, a free radical, maintaining the microcirculation and killing microbes. It should be noted that, despite many benefits of Arg supplementation, the excessive production of nitric oxide by inducible nitric oxide synthase can cause refractory hypotension and tissue injury. Moreover, Arg supplementation administered to patients with severe inflammation may exacerbate inflammatory responses. Therefore, appropriate timing and Arg dosage, as well as patient conditions for supplementation are the keys to clinical Arg use. It must be known that excessive doses should be avoided in patients with inflammatory status [37].

PUFAs modulate immune activity through two ways. Omega-3 fatty acids are metabolized to less inflammatory and less immunosuppressive eicosanoids than omega-6 fatty acids. It is known that omega-3 and omega-6 fatty acids are metabolized by the same enzymes, and therefore omega-3 fatty acids prevent excessive inflammatory responses and immunosuppression through competition for enzyme use [37]. Omega-3 fatty acids are also metabolized to anti-inflammatory mediators, such as resolvins and protectins, which rapidly terminate inflammation by modulating polymorphonuclear neutrophils (PMNs) and macrophage functions in the inflammatory site [37]. PUFAs increase the production of some prostaglandins (PGs) and leukotrienes, reducing the proinflammatory potential, and decrease the production of some other PGs (PGE2) and leukotrienes, reducing the cytotoxicity of macrophages, lymphocytes, and natural killer (NK) cells. Additionally, they decrease prostacyclin and thromboxane (TX)-A2 production and increase the antiaggregatory substance TXA3 [29]. It has been reported that PUFAs decrease excessive inflammatory responses, but they are not immunosuppressive, which is very important in cancer patients in the postoperative course [29,64,65]. Clinically, it has been observed that PUFAs decrease the infection rate and hospital stays in surgical and intensive care units patients. Pradelli et al. [62] reported that PUFAs had decreased inflammatory mediators, improved liver and lungs function, the level of antioxidants and composition of fatty acids in the phospholipid membrane [66]. One of the PUFAs, eicosapentaenoic acid (EPA), competes with the arachidonic acid (AA) for cyclooxygenase and 5-lipooxygenase binding sites, reduces the production of prostaglandine PGE2 from AA, and thereby reduces tissue inflammation [25]. Additionally, it reduces stress-induced immunosuppression after stressful surgery [24]. PUFAs reduce the production of prostacyclin and thromboxane (TX-A2), prostaglandin G2 and leukotrienes, and modulates cell mediated immunity [23]. Nucleotides are necessary for the proliferation of immune cells and the cells important for wound healing [29,67]. They regulate T-cell-dependent antibody production and lymphocyte function [36,68]. Formulas enriched with omega-3 fatty acids have normalized TPN-induced hepatic mononuclear cells (MNCs) dysfunction in animals. Clinically, PUFA are used to treat TPN-induced liver dysfunction [37].

## 7. Summary and Conclusions

In summary and conclusion, the reports regarding the role of IN in patients undergoing PD are non-numerous and contradictory. In spite of ERAS guidelines on clinical nutrition in surgery, currently there is no strong recommendation regarding IN in patients undergoing PD. Theoretically, immunonutrients significantly modulate the immune response. Therefore, they are useful for the recovery of patients undergoing major surgery, because for patients who are in the catabolism and have stress induced by the disease and surgical injury, the immune response is impaired and should be modulated in order to improve prognosis. Most authors recommend the use of preoperative oral IN in order to decrease the rate of the postoperative infectious rate and the length of hospital stay. Parenteral supplementation is much less recommended. In most publications reporting the benefits of IN in patients undergoing PD, the significant influence of IN on the rate of infectious complications and the length of hospital stay has been proven. Based on the literature review, we also recommend oral preoperative immunonutrition, because it is the better documented compared to other IN formulae (postoperative and parenteral). Patients for preoperative IN can be hospitalized 6 days before surgery, but they can also receive IN at home. The proper selection of candidates for IN among patients undergoing PD is also important, in order to achieve maximal benefits, without hospital costs increasing. According to most authors, the impact of IN on the other parameters of postoperative outcome (overall and non-infectious complications rate) is rather doubtful. However, in some authors’ opinions, IN does not have any benefits for patients undergoing PD. Therefore, more multi-center prospective randomized control trials are needed in order to establish the roles and indications for the administration of IN in patients undergoing PD.

## Figures and Tables

**Table 1 nutrients-12-02547-t001:** Search Strategy of the Literature Review.

1. All papers on IN in surgery and oncology in the PubMed database were found.
2. The articles on the role of immunonutrition in major gastrointestinal surgery were selected and reviewed.
3. Papers regarding immunonutrition in patients undergoing pancreaticoduodenectomy were selected and discussed.
4. Additionally, articles on characteristics of immunonutrients (glutamine, arginine, omega-3 fatty acids, nucleotides), in order to show the role of their role in surgical patients, were found, reviewed and cited.

**Table 2 nutrients-12-02547-t002:** Summary of Studies on Immunonutrition in Pancreaticoduodenectomy.

Reference (Year of Publication)	Patients No.	Study Design	Study Conclusions
Gianotti et al. [21] (2002)	212	Prospective randomized trial Enteral IN, enteral standard, parenteral	A lower rate of complications and a shorter length of hospital stay in IN group
Miyauchi et al. [22] (2019)	60	Prospective, randomized clinical trial Preoperative (1) and perioperative (2), oral/enteral IN	No differences between two groups
Silvestri et al. [23] (2016)	54	Case control, Preoperative IN vs. standard oral diet	No differences in mortality and morbidity rate A lower rate of infectious complications and a shorter length of hospital stay in IN group
Suzuki et al. [24] (2010)	30	Prospective randomized study Perioperative, postoperative, control	Reduction in infectious complications and immunosuppression in perioperative group
Aida et al. [25] (2013)	50	Prospective randomized study Preoperative oral IN and standard diet	A lower rate and severity of complications in preoperative IN Differences in levels of immune parameters between groups
Gade et al. [26] (2016)	35	Randomized controlled trial Preoperative oral and standard diet	No differences in complication rate and hospital stays between groups
Hamza et al. [27] (2015)	37	Randomized clinical trial Perioperative IN and standard diet	A favorable modulation of inflammatory response and improvement of systemic immunity in IN group
Ashida et al. [28] (2018)	20	Double-blinded randomized controlled trial Perioperative IN and standard diet	No impact of IN on hypercytokinemia and infectious complications
Shirakawa et al. [29] (2012)	25	Case control study Preoperative IN and standard diet	A lower rate of wound infection in IN group
Tumas et al. [30] (2020)	70	Prospective randomized study Preoperative IN and standard diet	No differences in overall complication rate Differences in severe/multiple complication rate A lower rate of severe/multiple complications in IN group
Guan et al. [17] (2019)	299	Meta-analysis of randomized of controlled trials Enteral IN and control	No differences in overall complication rate No differences in non-infectious rate A lower infectious complication rate and a shorter length of hospital stay in IN group No differences in mortality rate
Takagi et al. [16] (2020)	349	Systematic review of controlled trials Pre-, peri-, postoperative IN and control	A lower overall and infectious complications rate in IN group No differences in incidence of major complications mortality, POPF, DGE
Sungho et al. [32] (2006)	60	Prospective, randomized, double-blind, and controlled Clinical trial Parenteral Gln and control	No differences in complication rate and length of hospital stay, chemical and nutritional parameters
Furakawa et al. [31]	298	Retrospective cohort study Preoperative IN and standard diet High and low SMI groups	A lower infectious complications rate in IN group in high and low SMI groups OR seven times higher in low SMI than high SMI group

IN, immunonutrition; No., number; POPF, Postoperative Pancreatic Fistula; DGE, Delayed Gastric Emptying; Gln, Glutamine; SMI, Skeletal Muscle Mass Index, OR, Odds Ratio.

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
