# Peer review of "The Role of Immunonutrition in Patients Undergoing Pancreaticoduodenectomy"

_nutrients, 2020, doi:10.3390/nu12092547_

Round 1
Reviewer 1 Report
The authors present a well written review focusing on the role of immunonutrition in patients undergoing pancreaticoduodenectomy.
The review includes the most significant and updated articles on the topic.
Some comments as follows:
The authors should illustrate the search strategy of their literature review in a table.
- Introduction Line 25: ‘we are still looking...’. This sentence should be re-written in impersonal form.
Line 37 : ‘In the another study’ The authors are invited to check the manuscript in order to improve English form and grammar.
ERAS guidelines (11th, 12th, etc) should be appropriately referenced.
Lines 59-64: this part may be cancelled because the article is focused on PD and the authors have already declared that immmunonutrition in patients undergoing PD is a debated topic.
Line 66: it is not important to say that the article had been published in Pancreas. It is in the reference.
The paragraph “Immunomodulating mechanisms of nutrients used in IN” is too short and the topic should be better addressed. Some other articles can be taken into account, such as Andersson R et al . Immunomodulation in surgical practice. HPB (Oxford). 2006;8(2):116-123. doi:10.1080/13651820410016660, and others.
The content of paragraph 7 should be simply added to the paragraph 5, where the role of Glutamine is discussed.
Author Response
Dear Editor,
Thank you for peer reviewing of our manuscript nutrients-902887, entitled " The Role of Immunonutrition in Patients Undergoing Pancreaticoduodenectomy ".
Thank you for your questions and comments. We have fully addressed all the comments and my responses appear below. Our revised work includes corrections according to reviewers’ comments in the text. The changes, made according to reviewers’ comments, are highlighted in red print in the text.
We take this opportunity to express my gratitude to the reviewers for their constructive and useful remarks. Their comments allowed us to identify areas in my manuscript that needed modification.
We also thank you for allowing me to resubmit a revised copy of the manuscript.
We hope that the revised manuscript is now acceptable for publication in Nutrients.
Responses to Reviewer 1.
- The authors should illustrate the search strategy of their literature review in a table.
Answer:
The search strategy has been presented in Table 1. Reference to Table 1 was added in the text (Line 64-65) as follows:
The search strategy of the literature review (step by step) was presented in Table 1.
- Introduction Line 25: ‘we are still looking...’. This sentence should be re-written in impersonal form.
Answer:
The sentence (Line 25), We are still looking for ways to reduce the incidence of postoperative complications after PD, was re-written in impersonal form as follows:
The ways to reduce the incidence of postoperative complications after PD are still looking for.
- Line 37 : ‘In the another study’ The authors are invited to check the manuscript in order to improve English form and grammar.
Answer:
The sentence, (Line 37), In the another study, the authors demonstrated that patients with gastrointestinal cancers fed before and after surgery with a diet supplemented with arginine, omega-3 fatty acids and nucleotides, had a significant decrease in postoperative infectious complications and length of hospital stay in comparison to patients fed with a standard enteral formula. The decrease of morbidity rate and hospital stay can significantly reduce hospital costs [4,6], was re-written as follows:
In the other study, the authors demonstrated a significant decrease in postoperative infectious complications and length of hospital stay in patients with gastrointestinal cancers fed before and after surgery with a diet supplemented with arginine, omega-3 fatty acids and nucleotides, in comparison to patients fed with a standard enteral formula.
The manuscript has been checked and English form and grammar have been improved.
- ERAS guidelines (11th, 12th, etc) should be appropriately referenced.
Answer:
All ERAS guidelines (11th, 12th, etc) (Line 45-56) have been appropriately referenced and reference number was added for each recommendation, as follows:
At the beginning, the most recent ERAS guidelines on clinical nutrition in surgery do not strongly recommend IN in all patients undergoing major surgery (including PD). According to 11th recommendation [7], parenteral glutamine supplementation may be considered in patients who can not be fed adequately enterally and, therefore, require parenteral nutrition (PN) (grade of recommendation: B). Currently, there is no clear recommendation regarding the supplementation of oral glutamine [7]. According to 12th recommendation [7], postoperative parenteral nutrition including omega-3-fatty acids should be considered only in patients who can not be adequately fed enterally and, therefore, require parenteral nutrition (grade of recommendation: B) [7]. According to 13th recommendation [7], peri- or at least postoperative administration of specific formula enriched with immunonutrients (arginine, omega-3-fatty acids, ribo-nucleotides) should be given in malnourished patients undergoing major cancer surgery (grade of recommendation: B) [7]. Currently, there is no clear evidence for the use of these formulas enriched with immunonutrients vs. standard oral nutritional supplements exclusively in the preoperative period. Currently, no clear recommendation can be given regarding the parenteral or enteral supplementation of arginine as a single substance [7]. It should be emphasized that according to 17th strong (grade A) recommendation [7], preoperatively, oral nutritional supplements should be given to all malnourished cancer and high-risk patients undergoing major abdominal surgery; and moreover [7], 18th recommendation [7] prefers preoperative (5-7 days before surgery) administration of oral IN supplements including arginine, omega-3 fatty acids and nucleotides [7].
- Lines 59-64: this part may be cancelled because the article is focused on PD and the authors have already declared that immmunonutrition in patients undergoing PD is a debated topic.
Answer:
We have decided to include part regarding articles cosidering immunonutrition in other major gastrointestinal surgery in order to show that immunonutrition has an important role not only in patients undergoing pancreaticoduodenectomy, but also in other complex surgical procedures within the digestive tract. The comment is, that „this part may be cancelled“. Therefore, we understand the word “may” that the omission of this part of the text was suggested by the reviewer and not strongly ordered and consequently, the decision to remove a paragraph should be ours. We believe this paragraph enriches the work and is necessary because it gives a broad view of the place of IN in surgery.
- Line 66: it is not important to say that the article had been published in Pancreas. It is in the reference.
Answer:
We have removed the words: in Pancreas journal.
- The paragraph “Immunomodulating mechanisms of nutrients used in IN” is too short and the topic should be better addressed. Some other articles can be taken into account, such as Andersson R et al . Immunomodulation in surgical practice. HPB (Oxford). 2006;8(2):116-123. doi:10.1080/13651820410016660, and others.
Answer:
This paragraph has been extended significantly, according to reviewer’s sugestion, including the suggested article and some others on this topic. The references were re-numbered, as follows:
We would like to present the immunomodulating mechanisms of mentioned above nutrients. Most authors have reported a decreased risk of infectious complications in patients following surgery. In order to show the „antiinfectious“ role of IN, the mechanism of postoperative infections should be explained. In patients undergoing PD, bacterial flora from the gut, especially Enterococci and Escherichia coli, translocate into the mesenteric lymph nodes or blood, where the most postoperative infections are observed. Some factors in the perioperative period can facilitate this bacterial translocation, like reduction in jaundice, postoperative intestinal motility, the use of antibiotics resulting in small bowel bacterial overgrowth, deterioration of mucosal barrier due to malnutrition, intestinal manipulation, and parenteral nutrition. Therefore, immunonutrients should influence on immunological response in patients following surgery [29,36].
Stress caused by PD induces systemic inflammation with production of inflammatory mediators, such as: IL-1 beta, IL-6, TNF-alfa. Excessive production of these cytokines, especially IL-6, is associated with an increased risk of postoperative infectious complications following major abdominal surgery including PD. Multiple studies have shown a significant postoperative hypercytokinemia in patients following PD. In order to decrease the risk of postoperative infectious complications in patients undergoing PD, production of these cytokines should be decreased. IN may suppress postoperative hypercytokinemia and reduce a rate of infectious complications following PD, which is associated with a shorter duration of hospitalization [28]. It has been shown, that PD is one of the most stressful surgeries. The proof of this theory is that the level of plasma IL-6 after PD was higher than after gastric and colorectal surgery. Furthermore, stress-induced immunosuppression was greater after PD compared to gastric and colorectal surgery [22]. IN modulates the inflammatory response and production of inflammatory mediators in patients undergoing PD. Additionally, it favourably modulates the systemic and mucosal immunity [27]. It is important, that IN has got most benefits, when it is administered a minimum of 5 days before PD. It is associated with the fact, that preoperative IN for a minimum 5 days, is necessary to show its immunometabolic results following PD [25]. In patients undergoing PD, reduction of the host response and the immunity is noted. They are facilitated by low caloric intake and by intestinal bacterial translocation that are observed in patients following PD. The risk factors facilitating immunological disturbances are: alteration of postoperative intestinal motility and loss of mucosal barrier function. The impaired immunological activity leads to an increased risk of postoperative infectious complications (wound infection, pneumonia, infection of the urinary tract, an infected pancreatic fistula, enteritis, sepsis) and this is associated with a prolonged hospital stay. Prolonged duration of hospitalization is associated with increased hospital costs [22,23]. Surgery leads to a transient immunosuppression and potential alterations in gastrointestinal tract function. In patients following surgical injury, an excessive inflammatory response and paralysis of cell-mediated immunity may be observed. It can lead to postoperative infectious complications. The gastrointestinal tract has been called “the undrained abscess” of multiple organ failure. The postoperative gut barrier failure is associated with an increase in intestinal permeability and bacterial translocation. It leads to activation of immunocompetent cells within the gut wall and associated lymph nodes: mucosa-associated lymphoid tissue (MALT) and gut-associated lymphoid tissue (GALT). The combination of increased barrier permeability and gut inflammation may potentially result in remote organ dysfunction caused by interaction between polymorphonuclear leukocytes and the endothelial lining [37]. Obstructive jaundice, as a result of tumor infiltration or compression on the common bile duct due to a basic disease, is frequently noted in patients undergoing PD. It has been proven that it has been associated with an impaired immune function, including both the systemic and local defence. The gut barrier dysfunction and endotoxemia were noted in these patients. An increased risk of infectious and septic postoperative complications has been seen in patients with obstructive jaundice. Biliary obstruction results in an increase in intestinal permeability, upregulation of HLA-DR expression on enterocytes and GALT, suggesting an immune activation [37].
It has been reported that perioperative IN significantly reduced the risk of postoperative infectious complications by 50% [38]. It should be emphasized, that although WHO and Centers for Disease Control and Prevention guidelines for prevention of surgical site infection did not refer to the significance of nutrition therapy in previous versions, the 2016 WHO guidelines listed IN as a method contributing to the prevention of surgical site infection [39]. According to the 2nd recommendation, multiple nutrient-enhanced formulas can be used to prevent surgical site infections in adult patients undergoing major surgery. According to authors of these recommendations, a careful selection of candidates for nutritional support is needed, because the use of enhanced nutrition support is expensive and requires additional work for hospital staff [39].
The rout of IN is also very important, because it influences on host defence mechanism. Comparing enteral and parenteral routes, the enteral nutrition (EN) is the better than total parenteral nutrition (TPN) [38]. EN preserves and TPN reduces gut‐associated lymphoid tissue cell number, gut immunoglobulin A (IgA) level, respiratory tract IgA level, hepatic mononuclear cell number, resident macrophage number, and exudative neutrophil number [38]. EN significantly and TPN poorly influences on resistance against viruses and bacteria, intracellular signalling activation, cytokine production, and nuclear factor‐κB activation. It is important, that survival in portal bacteriemia and bacterial peritonitis is good for EN and poor for TPN [38].
Utsumi et al. [40] reported that preoperative nutritional assessment using the Controlling Nutritional Status (CONUT) Score may predict the pancreatic fistula after PD. CONUT score is a tool used to assess nutritional status. It takes into account the following laboratory parameters: serum albumin level (indicating protein reserve), total cholesterol level (indicating calorie depletion), and total lymphocyte count (indicating loss of immune defence caused by immune malnutrition). The authors concluded that patients with high CONUT score are at high risk for POPF. According to them, preoperative immunonutrition might help reduce the POPF risk in these patients through modulating impaired immunity in them [40]. It regards the clinically relevant POPF classified as grade B and C according to the International Study Group on Pancreatic Fistula classification [41]. POPF with an elevated inflammatory response observed in blood examination and following the intravenous antibiotic administration was defined as grade B POPF caused by infection. In the case of the organ failure occurring, a grade C POPF is recognized. Therefore, immunomodulating influences not only typical infectious complications, but also can decrease a risk of POPF – the common complication following PD (incidence 11.4-64.3% of PD) [40].
- The content of paragraph 7 should be simply added to the paragraph 5, where the role of Glutamine is discussed.
Answer:
The content of paragraph 7 has been added tot he paragraph 5 and references have been re-numbered.

Reviewer 2 Report
This review article is writing about role of Immunonutrition(IN) in Patients Undergoing Pancreaticoduodenectomy(PD). This study concluded that IN influence on the rate of infectious complications and the length of hospital stay, but more multi-center prospective randomized control trials are needed in order to precise role of IN for patients undergoing PD. From my point of view, there are some problems with this manuscript.
- In page 1, line 23, reference is needed about “However, perioperative mortality rate still ranges 0-5%”
- References to each ERAS guideline should also be mentioned.
- Looks like the wrong reference was put on page 3, line 80
- The third paragraph describes several studies of IN.
However, it is necessary to mention in more detail the method of each studies.
For example, it seems necessary to further comment on what was the definition of a standard enteral formula , when the postoperative enteral feeding started, and what was administered as IN formula. - In fifth paragraph, line 165 to 167, reference should be also mentioned.
- In 6th paragraph, line 181 to 182, reference should be also mentioned.
- In 7th paragraph, line 191 to 192, referece should be also mentioned.
- It has only basic explanation for immunonutrients such as glutamin, arginine PUFA, etc. It would be better if there was a specific comment on how it could work for patients after PD. The content mentioned in this manuscript is only for general care after surgery.
- There seems to be nothing special about the content of this manuscript. The conclusion itself that more multi-center prospective randomized control trials are needed does not appear to be characteristic.
Author Response
Dear Editor,
Thank you for peer reviewing of our manuscript nutrients-902887, entitled " The Role of Immunonutrition in Patients Undergoing Pancreaticoduodenectomy ".
Thank you for your questions and comments. We have fully addressed all the comments and my responses appear below. Our revised work includes corrections according to reviewers’ comments in the text. The changes, made according to reviewers’ comments, are highlighted in red print in the text.
We take this opportunity to express my gratitude to the reviewers for their constructive and useful remarks. Their comments allowed us to identify areas in my manuscript that needed modification.
We also thank you for allowing me to resubmit a revised copy of the manuscript.
We hope that the revised manuscript is now acceptable for publication in Nutrients.
Responses to Reviewer 2.
- In page 1, line 23, reference is needed about “However, perioperative mortality rate still ranges 0-5%”.
Answer:
Reference number [1] has been added as follows:
However, perioperative mortality rate still ranges 0-5% [1].
- References to each ERAS guideline should also be mentioned.
Answer:
References to each ERAS guideline have been added as follows:
At the beginning, the most recent ERAS guidelines on clinical nutrition in surgery do not strongly recommend IN in all patients undergoing major surgery (including PD). According to 11th recommendation [7], parenteral glutamine supplementation may be considered in patients who can not be fed adequately enterally and, therefore, require parenteral nutrition (PN) (grade of recommendation: B). Currently, there is no clear recommendation regarding the supplementation of oral glutamine [7]. According to 12th recommendation [7], postoperative parenteral nutrition including omega-3-fatty acids should be considered only in patients who can not be adequately fed enterally and, therefore, require parenteral nutrition (grade of recommendation: B) [7]. According to 13th recommendation [7], peri- or at least postoperative administration of specific formula enriched with immunonutrients (arginine, omega-3-fatty acids, ribo-nucleotides) should be given in malnourished patients undergoing major cancer surgery (grade of recommendation: B) [7]. Currently, there is no clear evidence for the use of these formulas enriched with immunonutrients vs. standard oral nutritional supplements exclusively in the preoperative period. Currently, no clear recommendation can be given regarding the parenteral or enteral supplementation of arginine as a single substance [7]. It should be emphasized that according to 17th strong (grade A) recommendation [7], preoperatively, oral nutritional supplements should be given to all malnourished cancer and high-risk patients undergoing major abdominal surgery; and moreover [7], 18th recommendation [7] prefers preoperative (5-7 days before surgery) administration of oral IN supplements including arginine, omega-3 fatty acids and nucleotides [7].
- Looks like the wrong reference was put on page 3, line 80.
Answer:
The reference number was corrected as follows:
The authors noted statistically a lower rate of infectious complications (22.9% vs 43.7%, p = 0.034) and shorter hospital stay (18.3 ± 6.8 days vs 21.7 ± 8.3, p = 0.035) in IN group [23].
- The third paragraph describes several studies of IN.
However, it is necessary to mention in more detail the method of each studies.
For example, it seems necessary to further comment on what was the definition of a standard enteral formula , when the postoperative enteral feeding started, and what was administered as IN formula.
Answer:
The study designs and protocols have been presented as follows:
One of the most important studies regarding the role of IN in patients undergoing PD is the study of Gianotti et al. [21] published in 2002. This is a prospective, randomized trial carried out in 212 patients undergoing PD. Patients were randomized to receive a standard enteral formula (standard group including 73 patients) or an enteral IN formula enriched with arginine, omega-3 fatty acids, and RNA (Impact; Novartis Nutrition) (immunonutrition group including 71 patients), or total parenteral nutrition (parenteral group including 68 patients). Administration of enteral diet started 6 hours after surgery at 10 ml/h rate. The velocity was progressively increased by 20 ml/day until reaching full nutritional goal (25 kcal/kg/day). Enteral or parenteral infusion was continued until the patient’s oral intake was approximately 800 kcal/d. The authors noted a significantly lower rate of postoperative complications and the shorter length of hospital stay in the IN group (33.8% / 15.1 days) compared to the standard (43.8% / 17.0 days) or parenteral group (58.8% / 18.8 days), respectively [21].
In 2019, Miyauchi et al. [22] published a prospective, randomized clinical trial. The authors compared perioperative and preoperative IN assessing cell-mediated immunity and the infection rate in patients following PD. The authors administered oral IN (arginine, ω-3 fatty acids, and dietary nucleotides) (Impact; Nestle Health) and enteral IN to 30 patients before and after surgery (perioperative group); 30 patients received the same enriched formula before surgery and standard enteral nutrition following surgery (preoperative group). The patients received oral supplementation (1000 kcal/d) for 5 days before surgery. A total of 1000 ml of oral Impact contained 12.8 g of arginine, 3.4 g of omega-3-fatty acids, and 1.29 g of nucleotides. All patients were hospitalized by 6 days before surgery. Postoperative enteral feeding with oral Impact in the perioperative group or standard formula in the preoperative group started about 12 hours after surgery. A gastrostomy catheter, placed into the jejunum intraoperatively, was used for enteral feeding in both groups. Enteral feeding was started on postoperative day 1 at 20ml/h and was increased progressively by 20 ml/day until the full nutritional goal (25 kcal/kg/day) was reached. All patients were allowed to drink water on postoperative day 2, and oral food intake was allowed from postoperative days 3-5 according to clinical conditions. The enteral feeding was continued until oral intake was approximately 800 kcal/day. The authors concluded that there were no additional effects of perioperative, compared with preoperative IN on postoperative immunity and infectious complications in patients following PD [22].
In 2016, Silvestri et al. [23] analyzed the role of preoperative IN on postoperative outcome in patients undergoing PD. In this study, 54 well nourished patients undergoing PD received preoperative oral IN (Impact; Nestle) (Impact composition: Arginine 1.8 g, omega-3-fatty acids 0.6 g, nucleotides 0.2 g) for 5 days before surgery at dose 750 ml/day (3 packs). From the 1st postoperative day, all patients were treated with total parenteral nutrition at dose 30 kcal/kg/d and 0.2-0.3 g of Nitrogen per kg/day. Oral diet was introduced progressively from the 5th postoperative day (in the absence of POPF). These patients were compared with a control group receiving preoperative standard oral diet. The authors demonstrated no statistical differences in mortality (2.1% in each groups) and overall morbidity rate (41.6% vs 47.9%) between both groups. The authors noted statistically a lower rate of infectious complications (22.9% vs 43.7%, p = 0.034) and shorter hospital stay (18.3 ± 6.8 days vs 21.7 ± 8.3, p = 0.035) in IN group [23].
There are some articles on mechanisms of IN on reducing infectious complications following surgery in the literature. Suzuki et al. [24], in a prospective randomized study, determined whether IN influenced the immunological parameters as follows: cell-mediated immunity, differentiation of T helper type 1 (Th1) and Th2 cells, interleukin (IL)-17-producing CD4(+) helper T (Th17) cell response, and infectious complication rate after PD. In this study, the authors divided patients undergoing PD into 3 groups counting 10 patients: the perioperative group received IN (Impact; Ajinomoto Pharma) for 5 days (1000 kcal/day) before surgery, which was prolonged after surgery by enteral infusion, the postoperative group received early postoperative enteral infusion of the same enriched formula with no artificial nutrition before surgery, the control group received total parenteral nutrition postoperatively. Enteral feeding started at 12-48 hours after surgery at 10 ml/h rate. The velocity was increased progressively by 20 ml/day until 25 kcal/kg/day was reached. Oral food intake was allowed on postoperative day 7. The authors concluded that perioperative IN had reduced the rate of infectious complications and stress-induced immunosuppression after PD. In the authors' opinion, the modulation of Th1/Th2 differentiation and Th17 response may play important roles in this immunologic effect [24].
Aida et al. [25], in a prospective randomized study, assessed the impact of preoperative IN on postoperative complications and the participation of prostaglandin E2 (PGE2) on T-cell differentiation in patients undergoing PD. The patients were divided into two groups according to nutritional intervention: 25 patients with preoperative IN (Impact; Ajinomoto Pharma) at dose 1000 kcal/day and 25 patients in the control group with standard nutrition administered for 5 days before surgery (2000 kcal/day). Both groups received the same postoperatibe nutrition. A gastrostomy catheter, inserted into the jejunum intraoperatively, was used for enteral feeding in both groups. Enteral nutrition was started on 1st postoperative day at dose 20 ml/h rate and it was increased progressively by 20 ml/day. Oral food intake was started on 5th postoperative day with decreasing gradually of enteral feeding. The study showed that infectious complication rate and severity of complications according to Clavien-Dindo classification were lesser in the IN group compared to the control group. Also, significant differences in levels of immunological parameters were noted between two groups. In the IN groups, mRNA expression levels of T-bet, serum eicosapentaenoic acid and eicosapentaenoic acid/arachidonic acid ratios were significantly greater and the levels of plasma prostaglandin E2 were significantly lesser compared to the control group [25].
Gade et al. [26] assessed the impact of oral IN (Oral Impact Powder; Nestle) used for 7 days before surgery for pancreatic cancer on postoperative complications and length of hospital stay in a randomized controlled trial on 35 patients including 19 (54%) patients with IN oral intervention. The intervention with IN was ended on the day before surgery. The IN group was compared with the control patients receiving the standard diet. There were not any statistical differences in postoperative complications and length of hospital stay between analyzed groups in this study [26].
In the randomized clinical trial performed by Hamza et al. [27], 37 patients, undergoing PD for periampullary cancer, were randomized, including 17 patients with IN intervention (Impact; Novartis) and 20 patients receiving the standard isocaloric isonitrogenous diet (Fresubin). Patients received 3 cartons (200 ml per carton) of either feed per day for 14 days before surgery. No patient received total parenteral nutrition in the postoperative period of the trial. Postoperatively, feeding was administered via nasojejunal tube within 24 hours after surgery at a rate of 25 ml/h with increasing rate gradually to achieve a rate of 25 kcal/kg/day by the 3rd or 4th postoperative day. The enteral feeding was continued a minimum of 7 days postoperatively. The patients were fed for 14 days preoperatively and 7 days postoperatively. The authors noted a favourable modulation of the inflammatory response and improvement of systemic immunity in the IN group [27].
Ashida et al. [28] analyzed the impact of IN on the incidence of hypercytokinemia after PD in a double-blinded randomized -controlled trial. The study included 20 patients: 11 patients received preoperative IN (enteral diets enriched in eicosapentaenoic acid (EPA)) (Prosure; Abbot Japan) in addition to 1200 kcal of regular food and 9 patients were fed using an isocaloric isonitrogenous standard nutrition (600 kcal/day) without EPA (Procure Z; Nisshin Oillio Group) for 7 days before surgery due to periampullary cancer. This study did not demonstrate the significant impact of preoperative IN on the rates of postoperative hypercytokinemia or infectious complications in 50 patients following PD [28].
Shirakawa et al. [29] analyzed the impact of preoperative oral IN on postoperative outcome in patients undergoing PD. The study involved 25 patients divided into two groups: 18 patients receiving oral IN (Impact; Ajinomoto Pharma) 5 days before surgery (750 ml/day: 3 packs of 250 ml per day) and 15 patients without IN received a standard oral diet. The authors reported a significantly lower rate of wound infection in patients receiving IN compared to the control group (0 vs. 30.8%,p=0.012) [29].
It is important to know, which patients may have benefits from IN in order to select a group of patients who can be recommended IN as standard. In 2020, Tumas et al. [30], 70 patients undergoing PD for pancreatic tumors randomized into IN vs. control groups and stratified according to final histological diagnosis and nutritional status. The IN group received 5 days of preoperative IN (L-arginine 6.04 g/day and polyunsaturated fat 4 g/day) in addition to the usual preoperative nutritional management. The control group received a routine preoperative nutritional management only. All patients received infusions 200 ml of 5% glucose solution in the morning of surgery. On postoperative day 1–3, patients received normocaloric enteral formula in an increasing dose and gradually replaced by oral nutrition on postoperative day 4–5 according to the clinical conditions. The authors assessed the impact of IN on the overall complication rate and the rate of severe and/or multiple complications in patients with pancreatic tumors stratified according to final histological diagnosis-patients with pancreatic ductal adenocarcinoma (PDAC) vs. other tumours-and nutritional status. In this study, there were no significant differences in the overall complication rates in IN vs. control, patients with malnutrition vs. no malnutrition, PDAC vs. other pancreatic tumors. However, significant differences in the rates of severe and/or multiple complications in IN vs. control groups and in PDAC patients divided according to IN were demonstrated [30].
Furakawa et al. [31] assessed the impact of oral IN on infectious complications in low skeletal muscle mass patients after PD, in a retrospective, consecutive cohort study. Skeletal muscle mass was assessed using preoperative computed tomography images in 298 consecutive patients who underwent PD. Risk factors for postoperative infectious complications and the impact of preoperative IN on low SMI patients undergoing PD were analyzed. Of the 298 patients, 91 patients received preoperative IN, containing omega-3 fatty acids, arginine, and nucleotides (oral Impact; Nestle Health Science), for 5 days before surgery, in addition to a 50% reduction in the amount of regular dietary intake (1000 kcal/day). A total of 1000 ml of oral Impact contained 2.0 g of EPA, 12.8 g of arginine, and 1.29 g of nucleotides. Patients, not receiving IN, consumed regular normocaloric diet before surgery (2000 kcal/day). All patients receiving preoperative IN were hospitalized for 6 day prior surgery. Postoperative enteral feeding, using standard formula, was started on 1st postoperative day via gastrostomy catheter at 20 ml/h rate and increased progressively by 20 ml/day, in order to achieve 25 kcal/kg/day. All patients were allowed to drink water on 2nd postoperative day. The authors compared impact of IN on infectious complications between two groups: high and low skeletal muscle mass index (SMI). In high SMI patients, the rate of postoperative infectious complications was significantly lower in those who received IN than in those who did not receive IN (31.9 vs. 46.1%, respectively; OR, 1.82; P = 0.045). Similar results were reported in low SMI patients (26.3 vs. 83.6%, respectively; OR, 14.31; P < 0.001), but Odds Ratio (OR) was significantly (7 times) higher in low vs. high SMI patients. The authors observed a significantly lower plasma IL-6 level in low SMI patients receiving IN compared to those who did not receive IN. Therefore, authors concluded, that there was a stronger association with reduced infectious complications in patients who have low SMI and receive IN. Additionally, authors analyzed the association between main pancreatic duct diameter and IN. This study revealed, that there was no association between main pancreatic duct diameter and IN. There was no influence of main pancreatic duct diameter on outcome of IN. This study has shown, that patients with low SMI are candidates for IN before PD. It is very important conclusion, because sarcopenia occurs in patients undergoing PD. It may be caused by ageing, cancer, malnutrition, sepsis, and some chronic diseases [31].
- In fifth paragraph, line 165 to 167, reference should be also mentioned.
Answer:
The reference [38] has been cited and added to the references list (references have been re-numbered) as follows:
- Kłęk S. Miejsce glutaminy w leczeniu żywieniowym: spojrzenie krytyczne w świetle medycyny opartej na faktach. Post Żyw Klin. 2013; 27(3): 21-24.
- In 6th paragraph, line 181 to 182, reference should be also mentioned.
Answer:
The reference number [29] has been added as follows:
PUFAs increase production of some prostaglandins (PGs) and leukotrienes, reducing the proinflammatory potential, and decrease production of some other PGs (PGE2) and leukotrienes, reducing the cytotoxicity of macrophages, lymphocytes, and natural killer (NK) cells. Additionally, they decrease prostacyclin and thromboxane (TX)-A2 production and increase the antiaggregatory substance TXA3 [29].
- In 7th paragraph, line 191 to 192 (currently 269-271), reference should be also mentioned. The
Answer:
The reference numbers have been added as follows:
The positive role of Gln on prognosis has been proven in critically ill patients [37,48-51]. However, data regarding the use of Gln in these patients are also contradictory [40,48-57], because not all studies have confirmed this theory. Some authors have proven that high doses of Gln are associated with an increased mortality rate in critically ill patients [53-57].
- It has only basic explanation for immunonutrients such as glutamin, arginine PUFA, etc. It would be better if there was a specific comment on how it could work for patients after PD. The content mentioned in this manuscript is only for general care after surgery.
Answer:
The cited articles on the role of Gln regard not only general care, but also PD. Additionally, other citations regarding benefits of immunonutrition, taking into account the specific benefits for patients after PD, were added as follows:
Line 198-242:
Stress caused by PD induces systemic inflammation with production of inflammatory mediators, such as: IL-1 beta, IL-6, TNF-alfa. Excessive production of these cytokines, especially IL-6, is associated with an increased risk of postoperative infectious complications following major abdominal surgery including PD. Multiple studies have shown a significant postoperative hypercytokinemia in patients following PD. In order to decrease the risk of postoperative infectious complications in patients undergoing PD, production of these cytokines should be decreased. IN may suppress postoperative hypercytokinemia and reduce a rate of infectious complications following PD, which is associated with a shorter duration of hospitalization [28]. It has been shown, that PD is one of the most stressful surgeries. The proof of this theory is that the level of plasma IL-6 after PD was higher than after gastric and colorectal surgery. Furthermore, stress-induced immunosuppression was greater after PD compared to gastric and colorectal surgery [22]. IN modulates the inflammatory response and production of inflammatory mediators in patients undergoing PD. Additionally, it favourably modulates the systemic and mucosal immunity [27]. It is important, that IN has got most benefits, when it is administered a minimum of 5 days before PD. It is associated with the fact, that preoperative IN for a minimum 5 days, is necessary to show its immunometabolic results following PD [25]. In patients undergoing PD, reduction of the host response and the immunity is noted. They are facilitated by low caloric intake and by intestinal bacterial translocation that are observed in patients following PD. The risk factors facilitating immunological disturbances are: alteration of postoperative intestinal motility and loss of mucosal barrier function. The impaired immunological activity leads to an increased risk of postoperative infectious complications (wound infection, pneumonia, infection of the urinary tract, an infected pancreatic fistula, enteritis, sepsis) and this is associated with a prolonged hospital stay. Prolonged duration of hospitalization is associated with increased hospital costs [22,23]. Surgery leads to a transient immunosuppression and potential alterations in gastrointestinal tract function. In patients following surgical injury, an excessive inflammatory response and paralysis of cell-mediated immunity may be observed. It can lead to postoperative infectious complications. The gastrointestinal tract has been called “the undrained abscess” of multiple organ failure. The postoperative gut barrier failure is associated with an increase in intestinal permeability and bacterial translocation. It leads to activation of immunocompetent cells within the gut wall and associated lymph nodes: mucosa-associated lymphoid tissue (MALT) and gut-associated lymphoid tissue (GALT). The combination of increased barrier permeability and gut inflammation may potentially result in remote organ dysfunction caused by interaction between polymorphonuclear leukocytes and the endothelial lining [37]. Obstructive jaundice, as a result of tumor infiltration or compression on the common bile duct due to a basic disease, is frequently noted in patients undergoing PD. It has been proven that it has been associated with an impaired immune function, including both the systemic and local defence. The gut barrier dysfunction and endotoxemia were noted in these patients. An increased risk of infectious and septic postoperative complications has been seen in patients with obstructive jaundice. Biliary obstruction results in an increase in intestinal permeability, upregulation of HLA-DR expression on enterocytes and GALT, suggesting an immune activation [37].
It has been reported that perioperative IN significantly reduced the risk of postoperative infectious complications by 50% [38]. It should be emphasized, that although WHO and Centers for Disease Control and Prevention guidelines for prevention of surgical site infection did not refer to the significance of nutrition therapy in previous versions, the 2016 WHO guidelines listed IN as a method contributing to the prevention of surgical site infection [39]. According to the 2nd recommendation, multiple nutrient-enhanced formulas can be used to prevent surgical site infections in adult patients undergoing major surgery. According to authors of these recommendations, a careful selection of candidates for nutritional support is needed, because the use of enhanced nutrition support is expensive and requires additional work for hospital staff [39].
The rout of IN is also very important, because it influences on host defence mechanism. Comparing enteral and parenteral routes, the enteral nutrition (EN) is the better than total parenteral nutrition (TPN) [38]. EN preserves and TPN reduces gut‐associated lymphoid tissue cell number, gut immunoglobulin A (IgA) level, respiratory tract IgA level, hepatic mononuclear cell number, resident macrophage number, and exudative neutrophil number [38]. EN significantly and TPN poorly influences on resistance against viruses and bacteria, intracellular signalling activation, cytokine production, and nuclear factor‐κB activation. It is important, that survival in portal bacteriemia and bacterial peritonitis is good for EN and poor for TPN [38].
Utsumi et al. [40] reported that preoperative nutritional assessment using the Controlling Nutritional Status (CONUT) Score may predict the pancreatic fistula after PD. CONUT score is a tool used to assess nutritional status. It takes into account the following laboratory parameters: serum albumin level (indicating protein reserve), total cholesterol level (indicating calorie depletion), and total lymphocyte count (indicating loss of immune defence caused by immune malnutrition). The authors concluded that patients with high CONUT score are at high risk for POPF. According to them, preoperative immunonutrition might help reduce the POPF risk in these patients through modulating impaired immunity in them [40]. It regards the clinically relevant POPF classified as grade B and C according to the International Study Group on Pancreatic Fistula classification [41]. POPF with an elevated inflammatory response observed in blood examination and following the intravenous antibiotic administration was defined as grade B POPF caused by infection. In the case of the organ failure occurring, a grade C POPF is recognized. Therefore, immunomodulating influences not only typical infectious complications, but also can decrease a risk of POPF – the common complication following PD (incidence 11.4-64.3% of PD) [40].
Line 247-251:
It is a preferred fuel for rapidly proliferating cells such as gut mucosal cells (enterocytes), lymphocytes, and neutrophils [37,38]. It also improves neutrophil, lymphocyte, and intestinal function. This amino acid also maintains a normal GALT function and respiratory immunity [37]. It is a material for synthesis of glutathione, a potent intrinsic antioxidant, and enhances heat shock protein expression [38]. Therefore, it is useful in order to modulate impaired immune response and dysfunction of intestinal barrier in patients following PD.
- There seems to be nothing special about the content of this manuscript. The conclusion itself that more multi-center prospective randomized control trials are needed does not appear to be characteristic.
Answer:
The aim of this paper was the review and summary of the literature regarding immunonutrition in pancreaticoduodenectomy. This is not an experimental study. Therefore, it is difficult to include special and innovative content. The conclusion follows from a literature review. We have included the additional conclusion (Line 203-204) as follows:
Based on literature review, we also recommend oral preoperative immunonutrition, because it is the better documented compared to other IN formulas (postoperative and parenteral). Patients for preoperative IN can be hospitalized 6 days before surgery, but they also receive IN at home. The proper selection of candidates for IN among patients undergoing PD is also important, in order to achieve maximal benefits, without hospital costs increasing.

Round 2
Reviewer 2 Report
I agree to there revised manuscript.